# An Alzheimer-associated TREM2 variant occurs at the ADAM cleavage site and affects shedding and phagocytic function

Kai Schlepckow[1], Gernot Kleinberger[1,2] (iD), Akio Fukumori[3,†], Regina Feederle[2,3,4], Stefan F Lichtenthaler[2,3,5,6], Harald Steiner[1,3] (iD) & Christian Haass[1,2,3,*] (iD)

## Abstract

Sequence variations occurring in the gene encoding the triggering receptor expressed on myeloid cells 2 (TREM2) support an essential function of microglia and innate immunity in the pathogenesis of Alzheimer's disease (AD) and other neurodegenerative disorders. TREM2 matures within the secretory pathway, and its ectodomain is shed on the plasma membrane. Missense mutations in the immunoglobulin (Ig)-like domain such as p.T66M and p.Y38C retain TREM2 within the endoplasmic reticulum and reduce shedding as well as TREM2-dependent phagocytosis. Using mass spectrometry, we have now determined the cleavage site of TREM2. TREM2 is shed by proteases of the ADAM (a disintegrin and metalloproteinase domain containing protein) family C-terminal to histidine 157, a position where an AD-associated coding variant has been discovered (p.H157Y) in the Han Chinese population. Opposite to the characterized mutations within the Ig-like domain, such as p.T66M and p.Y38C, the p.H157Y variant within the stalk region leads to enhanced shedding of TREM2. Elevated ectodomain shedding reduces cell surface full-length TREM2 and lowers TREM2-dependent phagocytosis. Therefore, two seemingly opposite cellular effects of TREM2 variants, namely reduced versus enhanced shedding, result in similar phenotypic outcomes by reducing cell surface TREM2.

**Keywords** Alzheimer's disease; neurodegeneration; phagocytosis; regulated intramembrane proteolysis; TREM2

**Subject Categories** Genetics, Gene Therapy & Genetic Disease; Immunology; Neuroscience

See also: **P Thornton** *et al* (October 2017)

## Introduction

Inflammation and activation of brain-resident immune cells are common hallmarks of numerous neurological disorders. A pivotal role of microgliosis has been recognized since a long time specifically in neurodegenerative disorders (Lyman *et al*, 2014; Villegas-Llerena *et al*, 2016). A central role of microglial function in disease pathogenesis is now further supported by the identification of sequence variants and mutations in the triggering receptor expressed on myeloid cells 2 (TREM2) that are associated with an increased risk for several neurodegenerative disorders such as Alzheimer's disease (AD), frontotemporal lobar degeneration (FTLD), Parkinson's disease, and FTLD-like syndrome (Guerreiro & Hardy, 2013; Guerreiro *et al*, 2013; Jonsson & Stefansson, 2013; Rayaprolu *et al*, 2013; Borroni *et al*, 2014; Cady *et al*, 2014; Cuyvers *et al*, 2014) and in a homozygous state cause Nasu–Hakola disease (Klunemann *et al*, 2005). In the brain, TREM2 is preferentially expressed in microglia and is functionally required for migration, cytokine release, phagocytosis, lipid sensing, ApoE binding, shielding of amyloid plaques, and microglia proliferation (Kleinberger *et al*, 2014; Atagi *et al*, 2015; Bailey *et al*, 2015; Colonna & Wang, 2016; Ulrich & Holtzman, 2016; Yeh *et al*, 2016; Yuan *et al*, 2016). TREM2 is a type-1 transmembrane protein that shuttles to the plasma membrane where it exerts its cell autonomous biological functions. TREM2 undergoes regulated intramembrane proteolysis (RIP) (Lichtenthaler *et al*, 2011; Wunderlich *et al*, 2013) (Fig 1A), which is initiated on the cell surface via shedding of full-length TREM2 by ADAM10 (a disintegrin and metalloproteinase domain containing protein) (Kleinberger *et al*, 2014). Shedding by ADAM10 results in the liberation of soluble TREM2 (sTREM2), which can be detected in human cerebrospinal fluid (CSF) (Kleinberger *et al*, 2014; Heslegrave *et al*, 2016; Piccio *et al*, 2016; Suarez-Calvet *et al*, 2016a,b). The membrane-retained C-terminal fragment (CTF) is

1 Biomedical Center (BMC), Biochemistry, Ludwig-Maximilians-Universität München, Munich, Germany
2 Munich Cluster for Systems Neurology (SyNergy), Munich, Germany
3 German Center for Neurodegenerative Diseases (DZNE) Munich, Munich, Germany
4 Helmholtz Center Munich, German Research Center for Environmental Health, Institute for Diabetes and Obesity, Core Facility Monoclonal Antibody Development, Neuherberg, Germany
5 Neuroproteomics, Klinikum Rechts der Isar, Technische Universität München, Munich, Germany
6 Institute for Advanced Study, Technische Universität München, Garching, Germany
*Corresponding author. Tel: +49 89 4400 46549; E-mail: christian.haass@mail03.med.uni-muenchen.de
†Present address: Department of Psychiatry, Osaka University Health Care Center, Toyonaka, Osaka, Japan

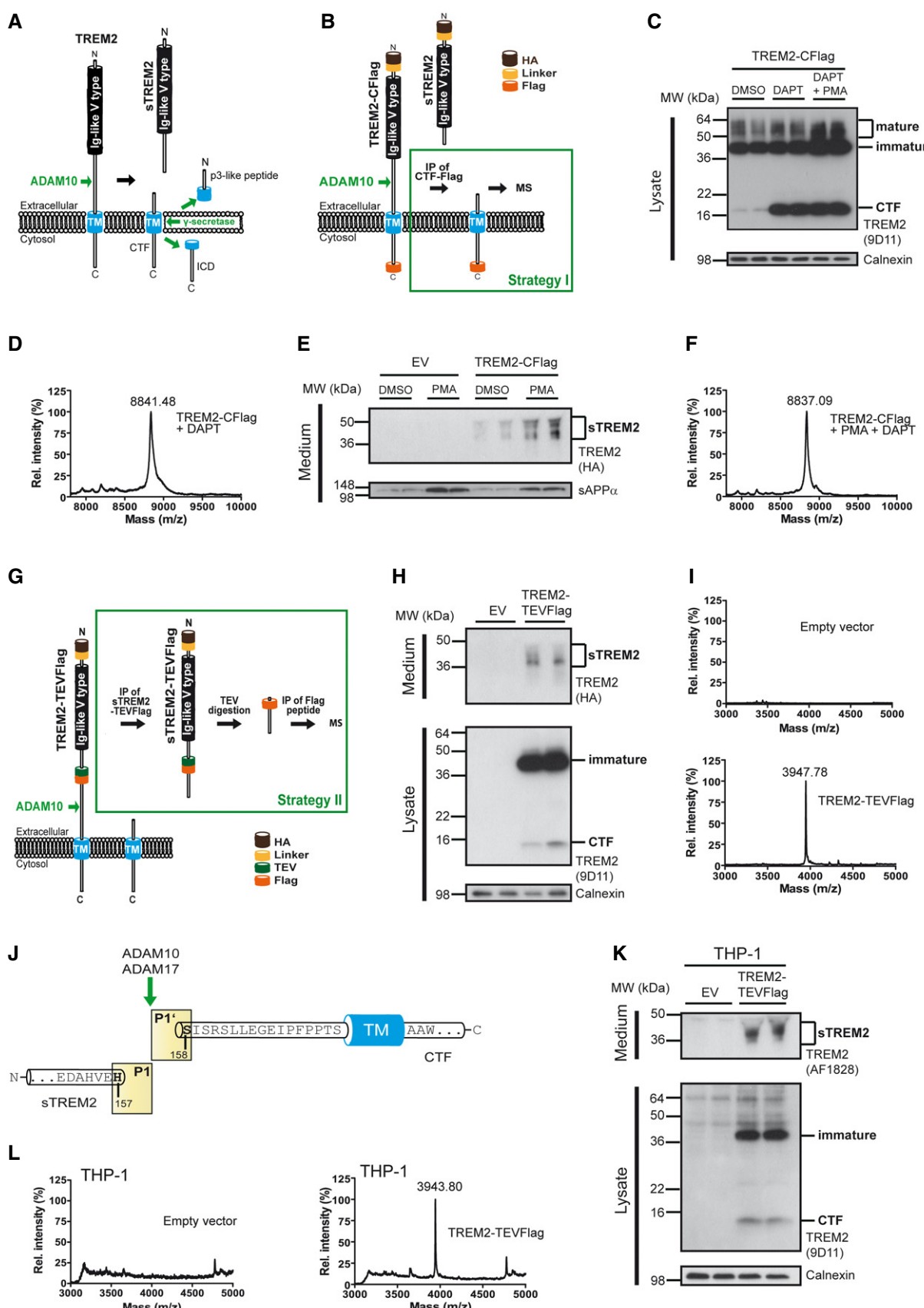

Figure 1.

**Figure 1.  TREM2 is shed at a single cleavage site in the ectodomain between histidine 157 and serine 158.**

A   Regulated intramembrane proteolysis of TREM2. ADAM10 initiates proteolytic processing of TREM2 by liberating its ectodomain (sTREM2). Subsequent processing of the membrane retained C-terminal fragment (CTF) by γ-secretase within the transmembrane domain (TM) releases the intracellular domain (ICD) into the cytosol. A short p3-like peptide (Haass *et al*, 1993) may be secreted.

B   Outline of strategy I for mass spectrometric (MS) determination of the N-terminal end of the TREM2 CTF enriched upon γ-secretase inhibition using DAPT.

C   Western blot analysis of TREM2 stably expressed in HEK293 Flp-In cells upon γ-secretase inhibition using DAPT. Application of DAPT leads to a robust accumulation of the TREM2 CTF under constitutive conditions as well as upon phorbol 12-myristate 13-acetate (PMA) mediated stimulation of TREM2 ectodomain shedding. The TREM2 9D11 antibody raised against the TREM2 C-terminus was used to detect TREM2, and calnexin levels were analyzed as a loading control. Here, as in all other Western blots in Figs 1 and 2, the two lanes represent samples from two separate wells seeded at the same time.

D   MALDI-TOF MS determination of the ectodomain cleavage site by immunoprecipitation of TREM2 CTF. The peak at 8,841.48 Da corresponds to a single cleavage site between histidine 157 and serine 158. Very minor additional peaks may represent cellular degradation products, as the N-terminal counterpart cannot be observed (see Fig 1I).

E   Stimulation of ectodomain shedding by PMA leads to strong increases in sTREM2 (anti-HA, upper panel) and sAPPα (2D8, lower panel). DMSO served as a vehicle control. EV, empty vector control.

F   MALDI-TOF MS determination of the ectodomain cleavage site upon ADAM17 stimulation using PMA. TREM2 CTFs were enriched by γ-secretase inhibition using DAPT (Fig 1C). The peak at 8,837.09 Da corresponds to a single cleavage site between histidine 157 and serine 158.

G   Outline of strategy II for MS determination of the C-terminal end of sTREM2 generated by ectodomain shedding.

H   Western blot analysis of full-length TREM2-TEVFlag (middle panel) and ectodomain shedding (upper panel) upon transient transfection of HEK293 Flp-In cells. EV, empty vector control.

I   MALDI-TOF MS determination of the ectodomain cleavage site following the strategy outlined in Fig 1G. The peak at a mass of 3,947.78 Da corresponds to cleavage C-terminal of histidine 157 and is absent in cells transfected with an empty expression vector.

J   TREM2 is shed predominantly C-terminal to histidine 157 as observed in two different cell types as well as under constitutive and PMA-stimulated conditions.

K   Western blot analysis of full-length TREM2-TEVFlag (middle panel) and ectodomain shedding (upper panel) upon transient transfection of THP-1 monocytes.

L   MALDI-TOF MS determination of the ectodomain cleavage site in THP-1 monocytes following the strategy outlined in Fig 1G. The peak at a mass of 3,943.80 Da corresponds to cleavage C-terminal of histidine 157 and is absent in cells transfected with an empty expression vector.

Source data are available online for this figure.

subsequently cleared via an intramembranous cleavage by γ-secretase (Fig 1A) (Wunderlich *et al*, 2013; Glebov *et al*, 2016). So far several mutations have been functionally investigated. Mutations within the immunoglobulin (Ig)-like domain such as p.T66M and p.Y38C apparently result in misfolding of TREM2 and retention of the immature protein within the endoplasmic reticulum (Kleinberger *et al*, 2014; Park *et al*, 2015; Song *et al*, 2017) although upon strong transient overexpression mutant proteins may escape retention (Kober *et al*, 2016). As a consequence of misfolding, reduced cell surface levels of TREM2 are observed and shedding is dramatically lowered leading to reduced sTREM2 and TREM2 CTF levels (Kleinberger *et al*, 2014). Consistent with that, a patient with a homozygous TREM2 p.T66M mutation had extremely low or even no detectable sTREM2 in the CSF (Kleinberger *et al*, 2014; Piccio *et al*, 2016). Lowered cell surface TREM2 results in reduced phagocytic activity (Kleinberger *et al*, 2014). Although initially discrepant results regarding the effects of a loss of TREM2 function on amyloid plaque pathology were reported (Jay *et al*, 2015; Wang *et al*, 2015), TREM2 loss of function may lead to the accumulation of fuzzy amyloid plaques suggesting a lack of phagocytic clearance of the plaque halo or reduced prevention of amyloid plaque growth (Wang *et al*, 2016; Yuan *et al*, 2016). In support of reduced phagocytic plaque degradation, we showed recently that immunotherapeutic clearance of amyloid plaques via phagocytosis is reduced in the absence of TREM2 (Xiang *et al*, 2016).

Most of the functionally investigated mutations are located within the Ig-like domain of TREM2. Misfolding of this domain, retention, and consequently reduced shedding appear to be a common readout of at least some of these variants. Although many sequence variants were found within the Ig-like domain, genetic studies also identified sequence variants within the stalk region and such mutants are unlikely to affect folding of the Ig-like domain. Since mutations in the stalk region may affect the efficacy and precision of ADAM-mediated shedding, we first determined the TREM2 cleavage site. Strikingly, ADAM-mediated cleavage within the stalk

region occurs C-terminal to histidine 157 exactly where the AD-associated variant p.H157Y is located (Guerreiro *et al*, 2013; Ma *et al*, 2014; Jiang *et al*, 2016; Song *et al*, 2017). Analysis of proteolytic processing of the mutant variant revealed that higher levels of sTREM2 are generated, a finding opposite to the reduced shedding observed for mutations within the Ig-like domain such as p.T66M and p.Y38C (Kleinberger *et al*, 2014). However, enhanced shedding of TREM2 p.H157Y leads to reduced cell surface full-length TREM2 and hence to reduced phagocytic activity. Thus, mutations located within the Ig-like domain or the stalk region reduce surface expression of TREM2 and probably its signaling activity via completely different cellular mechanisms.

## Results and Discussion

### TREM2 is cleaved by ADAM proteases between histidine 157 and serine 158

To determine the cleavage site of TREM2, we followed two independent approaches. First, we determined the N-terminus of the membrane retained CTF remaining after shedding of the full-length precursor (Kleinberger *et al*, 2014). To do so, we expressed C-terminally Flag-tagged TREM2 (TREM2-CFlag) in human kidney 293 cells (HEK 293) (strategy I; Fig 1B). We enriched for the ADAM generated TREM2 CTF by inhibiting its γ-secretase-mediated intramembrane cleavage with DAPT (Dovey *et al*, 2001). Consistent with previous results (Wunderlich *et al*, 2013), Western blot analysis revealed a massive accumulation of the CTF upon γ-secretase inhibition (Fig 1C). Immunoprecipitation followed by mass spectrometry of the DAPT enriched CTF revealed one major peak corresponding to a molecular mass of 8,841.48 Da (Fig 1D; Table 1). This corresponds to a CTF with an N-terminus at serine 158 (see also Fig 1J). Additional very minor peaks may be due to proteolytic degradation of the CTF most likely within lysosomes. This is

**Table 1.** Summary of identified peptides and comparison of observed masses to calculated masses. [M + H]⁺ indicates a singly charged peptide.

| Peptide | Cleavage | Sequence | Mass [M + H]$^+$ (Da) | |
|---|---|---|---|---|
| | | | Calculated | Observed |
| TREM2-CFlag WT (HEK) | N-terminal of serine 158 (P1′) | SISRSLLEGEIPF...DYKDDDDK | 8,840.08 | 8,841.48 |
| TREM2-CFlag WT (PMA, HEK) | N-terminal of serine 158 (P1′) | SISRSLLEGEIPF...DYKDDDDK | 8,840.08 | 8,837.09 |
| TREM2-CFlag p.H157Y (HEK) | N-terminal of serine 158 (P1′) | SISRSLLEGEIPF...DYKDDDDK | 8,840.08 | 8,832.76 |
| TREM2-TEVFlag (HEK) | C-terminal of histidine 157 (P1) | GDYKDDDDKLDHRDAGDLWFPGESESFEDAHVEH | 3,948.01 | 3,947.78 |
| TREM2-TEVFlag (THP-1) | C-terminal of histidine 157 (P1) | GDYKDDDDKLDHRDAGDLWFPGESESFEDAHVEH | 3,948.01 | 3,943.80 |

supported by the almost complete absence of such minor peaks in the analysis of the cleavage site of sTREM2 (see below).

So far ADAM10 has been described as the sole sheddase of TREM2 (Kleinberger *et al*, 2014). However, shedding is well known to be stimulated after protein kinase C activation. Under these conditions, shedding is predominantly performed by ADAM17 (Black, 2002; Saftig & Lichtenthaler, 2015). To prove whether TREM2 shedding can be stimulated similarly and whether cleavage still occurs at the same site, we treated HEK293 cells stably expressing TREM2-CFlag with phorbol 12-myristate 13-acetate (PMA). This resulted in a robust increase in TREM2 shedding (Fig 1E). Similarly, and in line with previous results (Nitsch *et al*, 1992), shedding of the amyloid precursor protein (APP) was also stimulated (Fig 1E; lower panel). To determine the cleavage site of TREM2 after PMA-stimulated shedding, we again enriched for the CTF by γ-secretase inhibition (Fig 1C) followed by immunoprecipitation with an anti-Flag antibody. Mass spectrometry of the CTF revealed one major peak corresponding to a molecular mass of 8,837.09 Da (Fig 1F; Table 1). Thus, after PMA-stimulated TREM2 shedding, the proteolytic cleavage also occurs between histidine 157 and serine 158 (see also Fig 1J).

To independently confirm the cleavage site, we introduced a TEV-protease cleavage site within the ectodomain of TREM2 after amino acid 132 followed by a Flag-tag (TREM2-TEVFlag; strategy II; Fig 1G) according to our previously described strategy for the identification of the cleavage site of presenilins (Fukumori *et al*, 2010). We first proved that such a modified TREM2 variant matures and undergoes shedding like unmodified TREM2. In cell lysates, full-length TREM2-TEVFlag was readily detected (Fig 1H; middle panel). HEK 293 cells expressing TREM2-TEVFlag also showed robust shedding of TREM2-TEVFlag (Fig 1H; upper panel), which is in line with the detection of the CTF in cell lysates (Fig 1H; middle panel). Next, sTREM2 was isolated by immunoprecipitation with an anti-Flag antibody and subsequently digested with TEV protease. Upon enrichment of the cleaved C-terminal peptide by anti-Flag immunoprecipitation (see Fig 1G), mass spectrometric analysis revealed a single sharp peak corresponding to a molecular mass of 3,947.78 Da, which was absent in conditioned media from HEK293 cells transfected with an empty expression vector (Fig 1I; Table 1). The observed molecular mass corresponds to a peptide terminating within the stalk region after histidine 157 (see also Fig 1J). Thus, both strategies independently identified one major cleavage site between histidine 157 and serine 158 (Fig 1J).

Next, we investigated whether shedding also occurs in a human monocytic cell line at the same site. To do so, we transiently transfected TREM2-TEVFlag into THP-1 cells (Tsuchiya *et al*, 1980).

TREM2-TEVFlag matures in THP-1 cells as in HEK293 cells and shedding of sTREM2 occurs as expected (Fig 1K). Following the same strategy as for HEK293 cells (see Fig 1G), mass spectrometric analysis revealed a single sharp peak corresponding to a molecular mass of 3,943.80 Da (Fig 1L; Table 1), which was absent in conditioned media from THP-1 cells transfected with an empty expression vector (Fig 1L; Table 1). The observed molecular mass corresponds to a peptide terminating within the stalk region after histidine 157 (see also Fig 1J), indicating the same TREM2 cleavage site in a human monocytic cell line.

## A disease-associated mutation at the cleavage site increases shedding

The cleavage site of TREM2 after histidine 157 coincides with the p.H157Y late-onset AD-associated mutation recently described in the Han Chinese population (Ma *et al*, 2014; Jiang *et al*, 2016) (Fig 2A). Since it is well known that an AD-related mutation at the site of β-secretase-mediated shedding of APP pathologically affects APP proteolysis (Citron *et al*, 1992; Cai *et al*, 1993; Haass *et al*, 1995b; Thinakaran *et al*, 1996), we investigated proteolytic processing of TREM2 p.H157Y. Surprisingly, and opposite to the p.T66M and p.Y38C mutations, we found enhanced shedding of the p.H157Y mutant by Western blotting (Fig 2B). This was independently confirmed by anti-HA ELISA-mediated quantitation (Fig 2C). Since shedding of p.H157Y is significantly increased and the amino acid sequence is changed at the P1 site, we next asked whether the cleavage still occurs after amino acid 157. As described above, we enriched for the CTF produced from TREM2 p.H157Y via inhibition of γ-secretase by DAPT (see also Fig 2H). Mass spectrometry of the enriched CTF revealed one major peak corresponding to a molecular mass of 8,832.76 Da, which is consistent with a predominant cleavage after amino acid 157 (Fig 2D and E; Table 1). Thus, TREM2 p.H157Y is cleaved at the very same position as wild-type (wt) TREM2. Furthermore, TREM2 p.H157Y is also shed by members of the ADAM family, since the protease inhibitors GM6001 and GI254023X significantly reduce shedding (Fig 2F). Enhanced shedding of the p.H157Y variant is accompanied by reduced levels of mature fully glycosylated TREM2 (Fig 2G). Despite increased shedding and reduced mature TREM2, we surprisingly observed less CTFs suggesting enhanced degradation (Fig 2G). In line with that, the CTF derived from TREM2 p.H157Y is recovered after inhibition of γ-secretase cleavage (Fig 2H). Furthermore, additional expression of DAP12, which forms a tight complex with TREM2 (Paradowska-Gorycka & Jurkowska, 2013), prevents γ-secretase-dependent degradation of the p.H157Y TREM2 CTF (Fig. 2I). Enhanced shedding

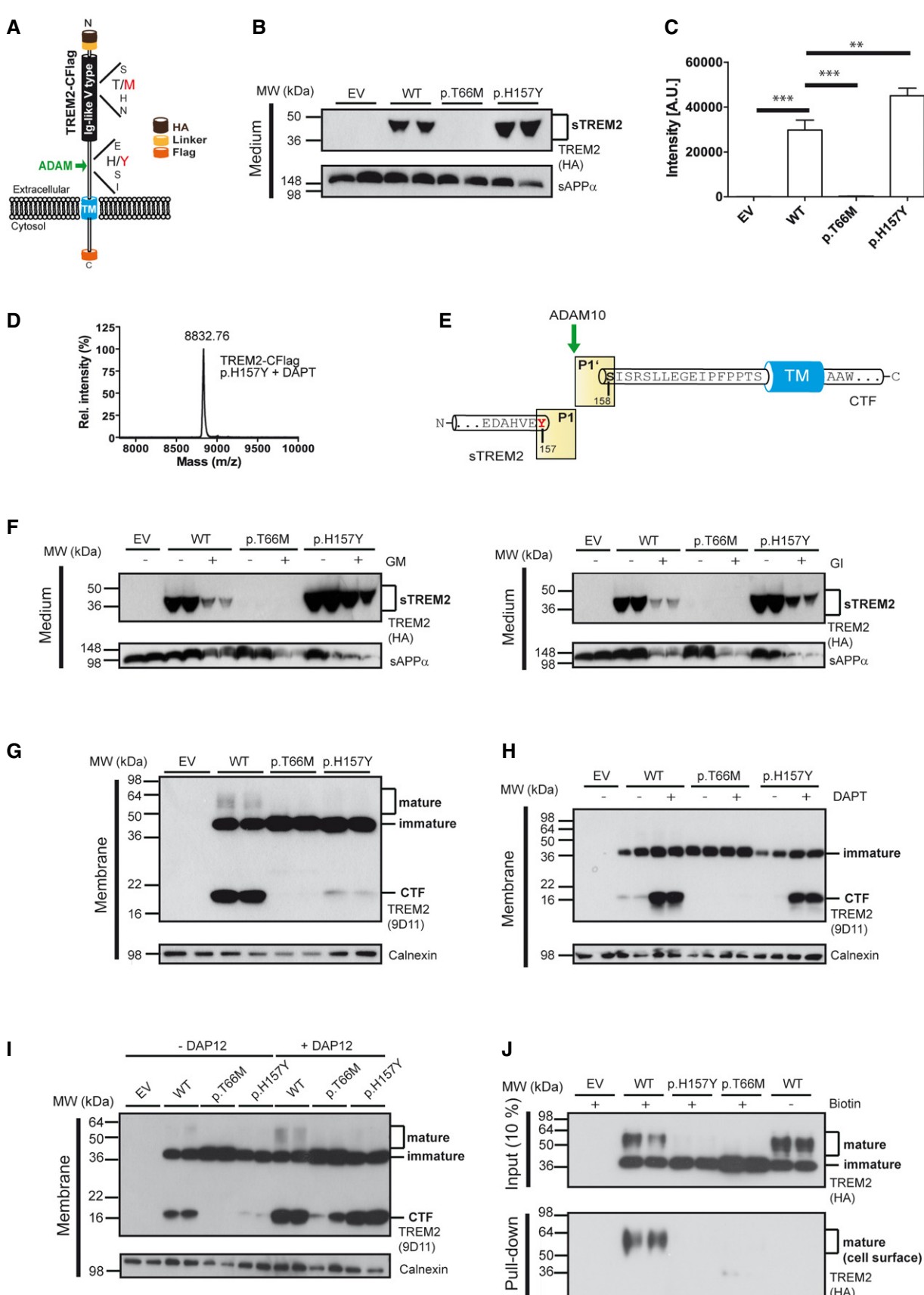

**Figure 2.**

**Figure 2.  A TREM2 sequence variant at the cleavage site leads to an increased shedding.**

A  The TREM2 variant p.H157Y is located at the P1 site where shedding occurs via proteases of the ADAM family. In addition, the location of the p.T66M mutation within the Ig-like V-type domain is indicated.

B  Increased shedding of TREM2 p.H157Y. Western blot analysis of conditioned media of HEK293 Flp-In cells stably transfected with cDNAs encoding HA and Flag-tagged wt TREM2 or TREM2 p.T66M and p.H157Y. Anti-HA antibody was used for detecting TREM2, and sAPPα levels were analyzed as a loading control. EV, empty vector control.

C  ELISA-mediated quantitation of sTREM2 (Suarez-Calvet *et al*, 2016a) in conditioned media from HEK293 cells stably expressing wt TREM2 or TREM2 p.T66M and p.H157Y. Error bars indicate SEM of seven independent experiments. One-way ANOVA (with Dunnett's *post hoc* test against wt) was used for statistical analysis; wt versus EV: \*\*\**P* < 0.0001; wt versus p.T66M: \*\*\**P* < 0.0001; wt versus p.H157Y: \*\**P* = 0.002.

D  MALDI-TOF MS analysis of the CTF derived from shedding of TREM2 p.H157Y identifies a single peak at a mass of 8,832.76 Da corresponding to a single cleavage site between tyrosine 157 and serine 158.

E  TREM2 p.H157Y is shed predominantly C-terminal to tyrosine 157.

F  Western blot analysis of TREM2 ectodomain shedding in the presence and absence of ADAM protease inhibitors. In line with previous findings (Kleinberger *et al*, 2014), the broad ADAM inhibitor (GM6001; left panel) and the more selective ADAM10 inhibitor (GI254023X; right panel) reduce shedding of wt TREM2, TREM2 p.H157Y as well as sAPPα.

G  Western blot analysis of membrane fractions of HEK293 Flp-In cells stably expressing wt TREM2, TREM2 p.T66M, or TREM2 p.H157Y. The TREM2 9D11 antibody raised against the TREM2 C-terminus was used to detect TREM2, and calnexin levels were analyzed as a loading control.

H  Western blot analysis of membrane fractions derived from cells stably expressing wt TREM2, TREM2 p.T66M, and TREM2 p.H157Y. γ-Secretase inhibition allows robust accumulation of the TREM2 CTF after shedding of TREM2 p.H157Y. Calnexin levels were analyzed as a loading control.

I  Western blot analysis of membrane fractions derived from cells stably expressing wt TREM2, TREM2 p.T66M, and TREM2 p.H157Y together with or without transiently expressed human DAP12. Note that co-expression of human DAP12 allows the recovery of large amounts of the TREM2 p.H157Y CTF. Calnexin levels were analyzed as a loading control.

J  Western blot analysis of cell surface biotinylated TREM2 reveals dramatically reduced levels of surface-exposed TREM2 p.H157Y, whereas robust amounts of cell surface wt TREM2 are detected. The anti-HA antibody was used to detect TREM2.

Source data are available online for this figure.

suggests reduced levels of TREM2 on the plasma membrane. Levels of cell surface mutant and wt TREM2 were determined by cell surface biotinylation. In line with our previous findings (Kleinberger *et al*, 2014), wt TREM2 was readily observed on the plasma membrane (Fig 2J). In contrast, p.H157Y could not be biotinylated on the cell surface similar to the previously investigated p.T66M mutant (Kleinberger *et al*, 2014) (Fig 2I). Thus, the histidine-to-tyrosine exchange at amino acid 157 increases shedding of mutant TREM2 and, as a consequence, reduces cell surface levels of the fully mature protein.

**The disease-associated mutation reduces phagocytic activity**

Both the p.T66M and the p.H157Y mutations lead to reduced cell surface TREM2, albeit via opposite cellular mechanisms. Since the amount of cell surface TREM2 correlates with its cell autonomous function (Kleinberger *et al*, 2014; Song *et al*, 2017), we hypothesized that both mutations may result in a similar loss of function. To investigate TREM2-mediated function, we analyzed TREM2-dependent phagocytosis. In line with our previous findings (Kleinberger *et al*, 2014), wt TREM2 readily promoted uptake of *Escherichia coli* conjugated to pHrodo (Fig 3). Uptake was specific since treatment with cytochalasin D blocked engulfment of *E. coli*. Strikingly, cells expressing TREM2 p.H157Y exhibited a significantly reduced phagocytic capacity (Fig 3), demonstrating that this mutation at least affects phagocytic uptake of bacteria.

  Sequence variations of TREM2 have been found in all domains of the protein. However, most of the mutations occur within the Ig-like domain. At least some of the mutant proteins, such as p.T66M and p.Y38C, are likely to be misfolded and retained within the endoplasmic reticulum (Kleinberger *et al*, 2014; Park *et al*, 2015; Song *et al*, 2017). Therefore, these types of mutations reduce cell surface TREM2 and consequently the release of sTREM2, as shedding predominantly occurs on the plasma membrane. Consistent with reduced cell surface TREM2, biological functions of TREM2 such as lipid sensing, ApoE binding, and phagocytosis are all decreased by

such mutations (Kleinberger *et al*, 2014; Atagi *et al*, 2015; Bailey *et al*, 2015; Yeh *et al*, 2016). Thus, for this class of mutations reduced release of sTREM2 may serve as a surrogate marker for

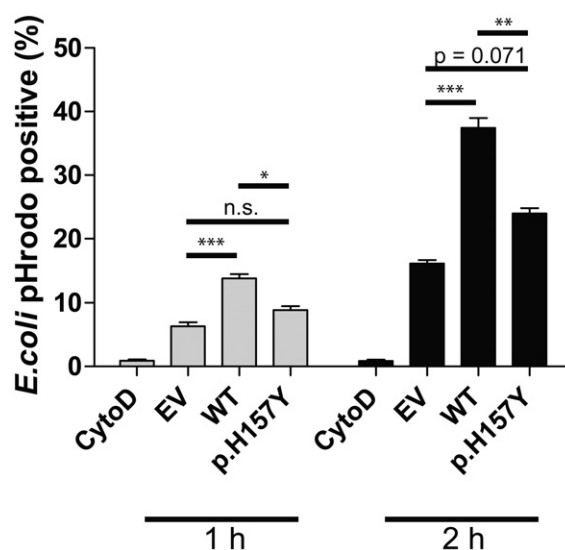

**Figure 3.  The TREM2 variant p.H157Y impairs phagocytosis.**
Phagocytic activity was measured using an *E. coli* pHrodo uptake assay. Cells were incubated with *E. coli* pHrodo particles for either 1 or 2 h (gray and black bars, respectively), and populations of phagocytically active cells were determined using flow cytometry. Cytochalasin D was used as a control. Data are shown as mean ± SD from four independent experiments ($n_{EV(1\ h)}$ = 11, $n_{WT(1\ h)}$ = 11, $n_{EV(2\ h)}$ = 12, $n_{WT(2\ h)}$ = 12) and three and two independent experiments for 1- and 2-h treatments, respectively ($n_{p.H157Y(1\ h)}$ = 8, $n_{p.H157Y(2\ h)}$ = 6). One-way ANOVA (with Tukey's *post hoc* test) was used for statistical analysis; EV versus WT (1 h): \*\*\**P* = 0.0005; EV versus p.H157Y (1 h): *P* = 0.413; WT versus p.H157Y (1 h): \**P* = 0.0141; EV versus WT (2 h): \*\*\**P* = 0.000014; EV versus p.H157Y (2 h): *P* = 0.071; WT versus p.H157Y (2 h): \*\**P* = 0.0031.

reduced biological function. In line with that, a patient with a homozygous p.T66M mutation showed almost no sTREM2 in plasma and cerebrospinal fluid (Kleinberger *et al*, 2014; Piccio *et al*, 2016), and this mutation is known to severely affect TREM2 functions (Kleinberger *et al*, 2014; Yeh *et al*, 2016). A number of sequence variants are now reported to occur within the stalk region of TREM2, which may not be involved in ligand binding and may also not be as sensitive to subtle structural changes as the highly folded Ig-like domain (Paradowska-Gorycka & Jurkowska, 2013). This region apparently serves as a spacer for the functionally important Ig-like domain but also contains the cleavage site recognized by the TREM2 sheddase that we have identified between histidine 157 and serine 158. Strikingly, a mutation, which increases the risk for late-onset AD, occurs in the Han Chinese population at P1 of the cleavage site (p.H157Y) (Ma *et al*, 2014; Jiang *et al*, 2016). In contrast to the above-described mutation, the p.H157Y mutation does not decrease but rather enhances shedding. On the first view, this may be a rather paradoxical finding. However, we also observed much less mature TREM2 p.H157Y, and consequently, the mutant protein could not be surface biotinylated similar to the TREM2 p.T66M mutation. Thus, enhanced shedding reduces the amount of full-length membrane-bound TREM2 on the cell surface. Apparently, the p.H157Y mutation provides a better substrate for shedding. As a consequence, shedding may already take place during transport of mutant TREM2 to the plasma membrane. This would be very similar to the Swedish APP mutation, which provides a better substrate for β-secretase-mediated processing and is therefore cleaved much earlier during cell surface transport than its wild-type counterpart (Haass *et al*, 1995b; Thinakaran *et al*, 1996). Moreover, ADAMs do not only cleave at the cell surface, but can also process their substrates (e.g. APP) already within the trans-Golgi (Haass *et al*, 1995a). For TREM2-mediated signaling, cell surface TREM2 bound to its co-receptor DAP12 is an absolute prerequisite (Colonna & Wang, 2016); thus, reduced cell surface TREM2 should result in reduced biological activity. This is indeed what we observed for the p.H157Y mutation. Similar to the p.T66M mutation (Kleinberger *et al*, 2014), this mutation interferes with phagocytosis and probably with all other surface TREM2-related functions as well. In line with that Song *et al* (2017) report, reduced ligand binding of p.H157Y and reduced phagocytic function may directly affect AD pathology. In that regard, it has been shown recently that in the absence of TREM2 fuzzy amyloid plaques accumulate, which may be a consequence of reduced engulfment of amyloid β-peptide by the dysfunctional microglia (Wang *et al*, 2016). Whether this is due to a direct inhibition of phagocytic activity and/or reduced migration to amyloid plaques remains to be shown. Moreover, similar to our recent analysis of mice endogenously expressing p.T66M (Kleinberger *et al*, 2017), the p.H157Y mutation should also be functionally investigated under an *in vivo* setting to confirm mutant TREM2 function in microglia in their natural environment.

Taken together, by locating a late-onset AD-associated TREM2 mutation right at P1 of the ADAM cleavage site, we demonstrate a novel cellular mechanism, which via enhanced shedding may lead to reduced function. Furthermore, our findings may open the opportunity to therapeutically modulate TREM2 function by selectively blocking access of ADAM proteases to the cleavage site between histidine 157 and serine 158.

# Materials and Methods

## cDNA constructs

cDNA constructs that were used in this study have been previously described (Kleinberger *et al*, 2014). Briefly, TREM2-CFlag as denoted in the main text is full-length TREM2 with N- and C-terminal HA and Flag tags, respectively. TREM2-DAP12 fusion constructs contained the ectodomain of TREM2 including aa169 (Proline169) fused to DAP12 (aa28–113). Furthermore, an amino acid change in the transmembrane domain of DAP12 from aspartic acid to alanine (p.D50A) was included (Hamerman *et al*, 2006). Additionally, the TREM2-DAP12 fusion constructs included a HA-tag and a linker sequence (SGGGGGLE) after the endogenous TREM2 signal peptide. The TREM2 missense mutation p.H157Y (CAC>TAC) was introduced into the respective plasmid by site-directed mutagenesis (Stratagene). The TREM2-TEVFlag construct was generated according to a previously reported strategy (Fukumori *et al*, 2010). The TEVFlag sequence (ENLYFQGDYKDDDDK) was introduced after amino acid 132. All constructs were verified by DNA sequencing (GATC Biotech).

## Cell culture

Generation and maintenance of stable HEK293 Flp-In cells lines were performed as previously described (Kleinberger *et al*, 2014). Human monocytic THP-1 cells were cultured in RPMI1640 medium supplemented with 10% FCS (Sigma), 1% penicillin/streptomycin, and 50 μM β-mercaptoethanol. THP-1 cells were kept in culture at densities between $2 \times 10^5$ and $1 \times 10^6$ cells per ml of cell culture medium, and transfections were performed at densities between $5 \times 10^5$ and $1 \times 10^6$ cells per ml of cell culture medium. Transfections were carried out with 1 μg of TREM2-TEVFlag DNA per $1 \times 10^6$ of THP-1 cells using a nucleofector device and the SG Cell Line 4D-Nucleofector® X Kit (both Lonza). GFP cDNA was transfected in parallel to check for transfection efficiency. Transient transfection of cells stably expressing wt or mutant TREM2 with human DAP12 was performed in a 6-well plate using 2 μg DNA/well and Lipofectamine 2000. If not stated otherwise, cell culture reagents were purchased from Thermo Fisher Scientific.

## Antibodies

For immunoblot detection, the following antibodies were used: rat monoclonal anti-HA conjugated to HRP (3F10; 1:700 to 1:2,000; Roche), goat polyclonal anti-human TREM2 (AF1828; 1:100; R&D Systems), rat monoclonal antibody against a C-terminal peptide of human TREM2 (9D11; 1:20; C-HGQKPGTHPPSELDCGHDPG), rat monoclonal antibody against sAPPα (2D8 (Shirotani *et al*, 2007); 1:100), and rabbit anti-calnexin (1:3,000, Enzo Life Sciences). The HRP-conjugated goat anti-rat (1:10,000; Santa Cruz Biotechnology), donkey anti-goat (1:10,000; Santa Cruz Biotechnology), and goat anti-rabbit IgG (1:10,000; Promega) were used as secondary antibodies.

## Cell surface biotinylation

Surface biotinylations were carried out as described previously (Kleinberger *et al*, 2014).

## Preparation of conditioned media, cell lysates, and immunoblotting

HEK293 Flp-In cells stably expressing respective cDNA constructs were seeded at a density of $1.5 \times 10^6/cm^2$, and medium was changed 48 h post-seeding. Inhibitors/activators used were GM 6001 (25 μM), GI 254023X (5 μM), DAPT (5 μM, all Sigma), and phorbol 12-myristate 13-acetate (PMA; 100 nM; Enzo Life Sciences). Conditioned medium was collected after 18–20 h except for PMA-treated cells where media were collected after 2 h. Membrane fractions were prepared as previously described (Kleinberger *et al*, 2014). Alternatively, total lysates were prepared using STE lysis buffer (150 mM NaCl, 50 mM Tris/HCl pH 7.6, 2 mM EDTA, 1% Triton X-100). Lysis was carried out on ice for 20 min, and lysates were obtained upon centrifugation at 15,871 *g* for 30 min at 4°C. To generate lysates upon transient transfection of THP-1 monocytes, cells were gently centrifuged (100 *g* for 10 min at 4°C) after overnight incubation at 37°C, washed once with ice-cold phosphate-buffered saline (PBS), centrifuged again (100 *g* for 10 min at 4°C), and subsequently lysed. Protein concentrations were measured using the BCA method, equal amounts of protein were mixed with Laemmli sample buffer supplemented with β-mercaptoethanol, separated by standard 15% SDS–PAGE, transferred onto polyvinylidene difluoride membranes (Hybond P; Amersham Biosciences), processed with respective antibodies, and developed using enhanced chemiluminescence technique (Pierce).

## Phagocytosis assay

Phagocytosis of fluorogenic *E. coli* particles (pHrodo Green, Molecular Probes) after 60- or 120-min incubation at 37°C was performed as described before (Kleinberger *et al*, 2014). As a negative control, phagocytosis was inhibited with cytochalasin D (10 μM; Sigma).

## sTREM2 ELISA

sTREM2 levels from conditioned media were essentially determined as previously described (Kleinberger *et al*, 2014). Rat anti-HA antibody (3F10; 1:1,000; Roche) was used as detection antibody.

## MALDI-TOF mass spectrometry analysis of ectodomain cleavage

For the identification of the N-terminus of the TREM2 CTF, HEK293 Flp-In cells stably expressing TREM2-CFlag (both wt and p.H157Y) were harvested in ice-cold PBS upon overnight treatment using DAPT. Upon treatment with PMA, cells were harvested 5 h post-treatment. Cell pellets were frozen at −20°C until use. Cells were lysed in lysis buffer (4% *n*-dodecyl β-D-maltoside, 0.1% *N*-octylglucoside, 10 mM Tris–HCl, pH 8.0, 5 mM EDTA, and 140 mM NaCl) containing protease inhibitor mix (Sigma) for 20 min on ice. Following a clarifying spin at 21,000 *g* for 10 min, supernatants were subjected to a second clarifying spin by ultracentrifugation at 100,000 *g* for 1 h and incubated with anti-FLAG M2-agarose (Sigma) overnight by rotation at 4°C. Beads were washed four times with immunoprecipitation/mass spectrometry (IP/MS) buffer (0.1% *N*-octylglucoside, 10 mM Tris–HCl, pH 8.0, 5 mM EDTA, and 140 mM NaCl) and two times with water. Beads were stored at −20°C until MS analysis. Control samples were generated using HEK293 Flp-In cells stably transfected with an empty expression vector (pcDNA5/FRT/TO, Thermo Fisher Scientific).

For the identification of the C-terminus of the shed ectodomain, the TREM2-TEVFlag cDNA construct was transfected transiently into HEK293 Flp-In cells followed by addition of fresh medium 24 h post-transfection. After further 5 h, the supernatant was collected and cleared by centrifugation at 15,000 *g* for 30 min at 4°C. The pH of the supernatant was adjusted to pH 8.0 using 1 M Tris/HCl (30 mM final concentration). 0.5 M EDTA pH 8.0 (3.5 mM final concentration) was added, and the supernatant was incubated with 40 μl anti-FLAG M2-agarose overnight by rotation at 4°C. Beads were washed four times with IP/MS buffer and two times with water. The TREM2 ectodomain was eluted from the beads with 40 μl 100 mM glycine pH 2.5 for 10 min by rotation at 4°C. Upon centrifugation (5 min at 1,200 *g*), the supernatant was neutralized by addition of 1/8 volume 1 M Tris–HCl pH 8.0. After addition of EDTA, DTT (final concentrations of 5 mM and 1 mM, respectively), and complete protease inhibitor (Roche), 10 units of AcTEV protease (Thermo Fisher Scientific) were added and digestion was carried out overnight at 4°C. Upon addition of 1 ml of IP/MS buffer, 10 μl anti-FLAG M2-agarose was added and immunoprecipitation was conducted for 1 h by rotation at 4°C. Beads were washed three times with IP/MS buffer and three times with water and stored at −20°C until MS analysis. In case of THP-1 monocytes, $30 \times 10^6$ cells were transiently transfected with the TREM2-TEVFlag cDNA construct. Upon overnight incubation at 37°C, cells were spun down and media were processed as described above to give samples for mass spectrometry. Control samples were generated using either HEK293 Flp-In cells or THP-1 cells transiently transfected with an empty expression vector (EV; pcDNA5/FRT/TO).

MS analysis was performed using a 4800 MALDI TOF/TOF analyzer (Applied Biosystems) essentially as described previously (Fukumori *et al*, 2010). Immunoprecipitated peptides were eluted from the beads using 0.3% trifluoroacetic acid (TFA) in 50% acetonitrile and saturated with α-cyano-4-hydroxy cinnamic acid. The dissolved samples were dried on a stainless plate and subjected to MALDI-TOF MS analysis.

## Statistics

The data were analyzed using GraphPad Prism 5 (GraphPad Software, Inc.) and in SPSS IBM, version 20.0. Group comparisons in the levels of sTREM2 (Fig 2C) or in phagocytosis assays (Fig 3) were analyzed by one-way ANOVA test followed by Dunnett's or Tukey's *post hoc* tests for pairwise comparisons, respectively. All tests were 2-tailed, and statistical significance was set to $P < 0.05$.

**Expanded View** for this article is available online.

## Acknowledgements

This work was supported by the Deutsche Forschungsgemeinschaft (DFG) within the framework of the Munich Cluster for Systems Neurology (EXC 1010 SyNergy), the DFG research unit FOR 2290 (Understanding intramembrane proteolysis), and the Cure Alzheimer's fund. This project has received funding from the Innovative Medicines Initiative 2 Joint Undertaking under grant agreement No 115976. This Joint Undertaking receives support from the European Union's Horizon 2020 research and innovation program and

## The paper explained

### Problem

A central role of microglial function in disease pathogenesis is strongly supported by the identification of sequence variants in the triggering receptor expressed on myeloid cells 2 (TREM2) that are associated with an increased risk for several neurodegenerative disorders. Most of the functionally investigated mutations are located within the Ig-like domain of TREM2. Misfolding of this domain, retention, and consequently reduced shedding appear to be a common read-out of at least some of these variants. Although many sequence variants were found within the Ig-like domain, genetic studies also identified sequence variants within the stalk region and such mutants are unlikely to affect folding of the Ig-like domain. Since mutations in the stalk region may rather affect the efficacy and precision of ADAM-mediated shedding, we determined the TREM2 cleavage site. Moreover, modulation of TREM2 shedding, for example, by its selective inhibition, may be an innovative therapeutic approach, for which knowledge of the exact cleavage site is absolutely required.

### Results

TREM2 is shed by proteases of the ADAM (a disintegrin and metalloproteinase domain containing protein) family C-terminal to histidine 157, a position where an AD-associated coding variant has been discovered (p.H157Y) in the Han Chinese population. Opposite to the characterized mutations within the Ig-like domain, such as p.T66M and p.Y38C, the p.H157Y variant within the stalk region leads to enhanced shedding of TREM2. Elevated ectodomain shedding reduces cell surface full-length TREM2 and lowers TREM2-dependent phagocytosis.

### Impact

Two seemingly opposite cellular effects of TREM2 variants, namely reduced versus enhanced shedding, result in similar phenotypic outcomes by reducing cell surface TREM2. Moreover, these findings may open the opportunity to therapeutically modulate TREM2 function by selectively blocking access of ADAM proteases to the cleavage site between histidine 157 and serine 158. In contrast to blocking ADAM 10/17 activity with conventional protease inhibitors, this approach would be selective and avoid cleavage inhibition of numerous ADAM10/17 substrates.

EFPIA. We would like to thank Axel Imhof for help with the mass spectrometry. Human monocytic THP-1 cells were kindly provided by Roche (Basel, Switzerland). We also thank Marc Suarez-Calvet for valuable help in statistical analyses.

## Author contributions

CH, KS, and GK designed the study and interpreted the results with additional help from HS and SFL. KS performed most experiments. Mass spectrometry was carried out by AF and KS. RF generated monoclonal antibodies. CH wrote the manuscript with input of all co-authors.

## Conflict of interest

C.H. is an advisor of F. Hoffmann-La Roche. C.H., G.K, and K.S. filed a patent (EP16180844) for the use of TREM2 cleavage modulators.

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
