## [Review Process File · EMBO Molecular Medicine]

An Alzheimer associated TREM2 variant occurs at the ADAM cleavage site and affects shedding and phagocytic function

Kai Schlepckow, Gernot Kleinberger, Akio Fukumori, Regina Feederle, Stefan F. Lichtenthaler, Harald Steiner & Christian Haass

Corresponding author: Christian Haass, Ludwig-Maximilians-University and DZNE, Germany

Review timeline:

Submission date:	07 February 2017
Editorial Decision:	15 March 2017
Revision received:	26 June 2017
Accepted:	14 July 2017

Transaction Report:

Editor: Céline Carret

1st Editorial Decision

15 March 2017

Thank you for the submission of your manuscript to EMBO Molecular Medicine. We have now heard back from the three referees who we asked to evaluate your manuscript.

You will see that all 3 are enthusiastic about the data and are supportive of publication pending minor revisions. We also would like to raise your attention to some needed editorial amendments.

1) Please address the minor comments of all referees. Please provide a letter INCLUDING the reviewer's reports and your detailed responses to their comments (as Word file).

2) Source Data:

As you know, we now encourage the publication of source data, particularly for electrophoretic gels, blots, but also microscopy images with the aim of making primary data more accessible and transparent to the reader. Would you be willing to provide a PDF file per figure that contains the original, uncropped and unprocessed scans of all or key gels used in the figure? The PDF files should be labeled with the appropriate figure/panel number (1 file/figure), and should have molecular weight markers; further annotation may be useful but is not essential. The PDF files will be published online with the article as supplementary "Source Data" files.

I look forward to seeing a revised form of your manuscript as soon as possible.

***** Reviewer's comments *****

Referee #1 (Remarks):

In this manuscript by Schlepckow et al., the authors investigate the effects of TREM2 variants associated with Alzheimer's disease. The authors found that the H157Y mutation increases shedding of mutant TREM2, and reduced phagocytic activity. The authors wrote a clear introduction, and performed their experiments rigorously. Therefore, this research manuscript is suitable for publication in EMBO after addressing the following comments:

Minor comments

1. The authors did not clarify why TREM2 p.H157Y leads to enhanced degradation of CTF by gamma-secretase cleavage.
2. How does the phagocytic function of WT TREM2 compare to mutant TREM2 (p.H157Y) in vivo? Has this been investigated by the authors or other groups? This can be discussed to address of the biological consequences.
3. Figure legends for figures 1-3 should include description of all abbreviations used in the figure
4. Fig. 1C, E, H, K should specify in the legend or Figure what the two lanes for each condition represent. Do the two lanes represent independent experiments?
5. Fig. 1K: in the upper panel, sTREM2 in both lanes of TREM2-TEVFlag are not on the same height on the blot. Please clarify.

Referee #2 (Remarks):

The manuscript by Schlepckow et al. presents a careful characterization of the effect of different mutations in Triggering Receptor expressed on Myeloid Cells 2 (TREM2) on its surface levels and ability to function as a receptor involved in phagocytosis. The authors characterize the cleavage site of TREM2 for ADAM10 and ADAM17, finding it to be identical for these two principal sheddases. Moreover, they compare and contrast the effect of a mutation in the Ig-like domain of TREM2 (T66M) on its cell surface expression with that of a mutation at the cleavage site (H157Y) that has been implicated in Alzheimer's disease. Interestingly, both types of mutations clearly decrease surface levels of full length TREM2, but by very different mechanisms. The mutation in the Ig-like domain lead to intracellular retention, whereas the H157Y mutation close to the cleavage site increases shedding, thus also lowering cell surface levels. The authors employ a phagocytosis assay with labeled *E. coli* to provide functional evidence that the H157Y mutation lower phagocytosis, suggesting that this mutation could also affect phagocytosis of Abeta plaques. Thus, they provide a compelling explanation for how the H157Y mutation might contribute to AD.

Overall, the data are of the highest quality, with excellent Western blots and mass spectrometric data and a convincing experimental approach. The analysis of *E. coli* endocytosis rounds out the study in a way that points to the functional relevance of the H157Y mutation. The introduction is well written and provides a good overview of the current status of the field, including previous work implicating ADAM10 as a key sheddase for TREM2. The discussion is supported by the results, and provides an appropriately balanced interpretation of these interesting findings. This is a timely contribution to the understanding of the role of TREM2 in AD, which should be of significant interest to the readers of EMBO MM.

Critique:

This reviewer has no concerns regarding the data or their interpretation, but feels that this already excellent manuscript could be further improved by performing the phagocytosis assay shown in figure 3 with the T66M mutant in the Ig-like domain to provide a side-by-side comparison. The outcome seems quite predictable, but would nevertheless be interesting to include.

Referee #3 (Comments on Novelty/Model System):

The present manuscript describes the identification of the cleavage site of TREM2, which very interestingly, corresponds the location of a known pathogenic mutant, p.H157Y. This finding is highly novel as it represents the first report to identify the TREM2 cleavage site. Given the known

impact of this and other TREM2 mutations in the pathogenesis of 'sporadic' AD, this information is timely and highly medically relevant.

The authors were thorough in their methodology used to definitely determine the cleavage site, employing multiple strategies to confirm their finding. Similarly, they utilized multiple cell systems to bolster the adequacy of their model system, which is appreciated.

Referee #3 (Remarks):

This manuscript serves as an important addition to previous findings from this group relating to TREM2 processing and the impact of pathogenic mutations on TREM2 processing and cell function.

The manuscript is clearly written and likewise, data are clearly presented in the corresponding figures. Methods and statistics are sufficiently described and experimental rigor is also sufficient in that multiple replicates are represented in the figures.

For all of the above reasons, this manuscript is suitable for publication with consideration to the following minor revision:

The authors end the manuscript with a demonstration of functional effects of the p.H157Y mutation on TREM2 relating specifically to phagocytosis. While the attempt to demonstrate the functional consequences of this mutation is appreciated, and this reviewer does not feel that functional experiment is necessary for publication, the experiment presented fails to sufficiently address the question asked. If included in the manuscript, the following considerations should be addressed:

Microglia cells are highly influenced by micro-environmental conditions and their functional/phagocytic response is highly context and stimulus specific. For that reason, one should take care when extrapolating in vitro findings from cell lines to the in vivo setting. Along those lines, it is important to mimic relevant conditions as much as possible. In this case, phagocytosis of Abeta would be much more relevant and appropriate than E coli and should be included since phagocytosis of E coli may not employ the same phagocytic mechanisms as Abeta. Abeta phagocytosis has been demonstrated by this group in past experiments, demonstrating their ability to conduct such an experiment.

Furthermore, microglial phagocytosis of amyloid plaques in the human AD brain (in the absence of additional stimulation, e.g. vaccination) is highly controversial and poorly demonstrated and understood thus far, both in terms of the occurrence and impact on disease pathogenesis. For that reason, the significance of the functional (phagocytosis) data presented (in p.H157Y mutants) on the pathogenesis of AD should be carefully interpreted and described given the high level of public interest in this topic to responsibly ensure appropriate understanding of these findings by non-specialists.

Overall, this is a very nice paper that I believe EMOBO readers will appreciate and from which they will derive benefit.

1st Revision - authors' response

26 June 2017

Referee #1

The authors did not clarify why TREM2 p.H157Y leads to enhanced degradation of CTF by gamma-secretase cleavage.

We agree with the referee that this is a major open question, specifically since in the accompanying paper the CTF was readily detected. When addressing this point we thought that TREM2 always forms a tight complex with DAP12 in myeloid cells. However, in HEK293, which we used to express ectopic TREM2 no DAP12 is expressed. Dimer formation is likely to abolish gamma-secretase from cleaving, since its substrate binding site as well as its catalytically active site apparently accepts monomeric substrates only. Thus, TREM2/DAP12 dimer formation may lead to the inhibition of gamma-secretase mediated clearance of TREM2 and therefore to stabilization of

TREM2 CTFs. Upon co-expression of TREM2 and DAP12 we indeed found a robust stabilization of TREM2 CTFs including the TREM2 H157Y CTF. This is now shown in the new figure 2I. Moreover, since TREM2 H157Y is a better substrate for ADAM cleavage, we assume in analogy to Swedish mutant APP (Haass et al., Nature Medicine, 1995), that monomeric TREM2 H157Y is cleaved earlier during the transport through the secretory pathway and then directly targeted to endosomes/lysosomes. This is now discussed accordingly.

How does the phagocytic function of WT TREM2 compare to mutant TREM2 (p.H157Y) in vivo? Has this been investigated by the authors or other groups? This can be discussed to address of the biological consequences.

We currently do not know this. However, in our recent amyloid seeding experiments in the TREM2 KO and the TREM2 T66M mouse, we observed enhanced seeding and higher coverage of the amyloid plaque area indicating a TREM2 dependent phagocytic clearance of amyloid plaques. Unfortunately, the H157Y mutation has not yet been introduced into the mouse genome. Nevertheless, so far we see a good correlation between in vitro and in vivo phenotypes of the T66M mutation (see also our recent publication by Kleinberger et al., EMBO J, 2017) and would expect similar parallels in H157Y mice. This is now discussed accordingly.

Figure legends for figures 1-3 should include description of all abbreviations used in the figure.

Abbreviations are explained in the main text, when used the first time.

Fig.1C, E, H, K should specify in the legend or Figure what the two lanes for each condition represent. Do the two lanes represent independent experiments?

The two lanes represent samples from two separate wells seeded at the same time. This is now mentioned in the Fig. 1 legend.

Fig.1K: in the upper panel, sTREM2 in both lanes of TREM2-TEVFlag are not on the same height on the blot. Please clarify.

The supernatants of cultured cells contain large amounts of serum proteins, which slightly affect protein migration. Variability in the apparent migration pattern are therefore not due to different molecular weights.

Referee #2

This reviewer has no concerns regarding the data or their interpretation, but feels that this already excellent manuscript could be further improved by performing the phagocytosis assay shown in figure 3 with the T66M mutant in the Ig-like domain to provide a side-by-side comparison.

We showed in our two previous publications that TREM2 T66M inhibits phagocytosis. We first demonstrated that the T66M mutation impairs phagocytosis in HEK293 overexpressing T66M mutant TREM2 (Kleinberger et al., Science Translational Medicine, 2014). More recently we confirmed this finding in mice where we introduced the T66M mutation via CRISPR/Cas9 genome editing (Kleinberger et al., EMBO J, 2017). Since T66M is an almost complete loss of function mutation, the effects on phagocytosis are rather pronounced.

Referee #3

For that reason, one should take care when extrapolating in vitro findings from cell lines to the in vivo setting. Along those lines, it is important to mimic relevant conditions as much as possible. In this case, phagocytosis of Abeta would be much more relevant and appropriate than E coli and should be included since phagocytosis of E coli may not employ the same phagocytic mechanisms as Abeta.

For in vivo analysis one must introduce the H157Y mutation into the mouse genome. We are about to do this, however, generation and phenotypic analyses of this mouse mutant will take a substantial amount of time and is therefore part of a different project. However, we want to emphasize that in

our recent publication (Kleinberger et al., *EMBO J*, 2017) we demonstrated a reduction of Aβ uptake in bone marrow derived macrophages carrying the T66M mutation. We are now pointing out in the discussion that functional implications of our study must be confirmed in an *in vivo* setting, at best in a CRISPR/Cas9 edited mouse model expressing endogenous TREM2 H157Y.

Corresponding Author Name: Christian Haas

Manuscript Number: EMM-2017-07672